# Can We Extend the Indications for Multilevel Surgery to Non-Ambulatory Children with Neuromuscular Diseases? A Safety and Efficacy Study

María Galán-Olleros [1,*], Ignacio Martínez-Caballero [1], Gonzalo Chorbadjian-Alonso [2,3], Rosa M. Egea-Gámez [4], David Sánchez-López [5], Ana Ramírez-Barragán [1], Manuel Fraga-Collarte [1] and Sergio Lerma-Lara [6,7]

1   Neuro-Orthopaedic Unit, Orthopaedic Surgery and Traumatology Department, Hospital Infantil Universitario Niño Jesús, 28009 Madrid, Spain
2   Unidad de Traumatología Infantil—Programa de Neuroortopedia, Departamento de Traumatología, Clínica Alemana de Santiago, Facultad de Medicina, Universidad del Desarrollo, Santiago 7650568, Chile
3   Servicio de Traumatología Infantil, Hospital Clínico San Borja Arriarán, Santiago 8360160, Chile
4   Spine Unit, Orthopaedic Surgery and Traumatology Department, Hospital Infantil Universitario Niño Jesús, 28009 Madrid, Spain
5   Orthopaedic Surgery and Traumatology Department, Hospital General La Mancha Centro, 13600 Alcázar de San Juan, Spain
6   Physiotherapy Departament, Centro Superior de Estudios Universitarios La Salle, Universidad Autónoma de Madrid, 28049 Madrid, Spain
7   Motion in Brains Research Group, Centro Superior de Estudios Universitarios La Salle, Universidad Autónoma de Madrid, 28049 Madrid, Spain
*   Correspondence: mgalanolleros@gmail.com

**Abstract:** A retrospective study that aims to analyze the safety and efficacy of single-event multilevel surgery (SEMLS) involving bifocal femoral osteotomy (BFO) performed in nine non-ambulatory children with neuromuscular diseases (NMD), with a median age of 12.86 years, to resolve both hip subluxation and ipsilateral knee flexion contracture that impaired standing, and to evaluate patient and caregiver satisfaction. Results: Median surgical time was 4 h 15 min (2 h 35 min–5 h 50 min). Hip flexion deformity improved by a median of 30° (15–35), while median improvement in knee flexion deformity was 30° (20–50). Only two patients could use a standing frame prior to surgery, although with increasing difficulty, while all children could use it following SEMLS. Mean follow-up was 27.47 months (24.33–46.9). Significant blood loss requiring transfusion was the only complication recorded (8/9). All caregivers reported slight, moderate, or significant improvement in all domains of the questionnaire, and all would undergo the procedure again and recommend it to others, as nearly all (8/9) were very satisfied. Conclusion: The findings of this study suggest that SEMLS including BFOs in non-ambulatory children with NMD can correct hip, knee, and foot deformities and simultaneously realign lower limbs to restore functional standing and wheelchair transfer. The functional results, safety, and degree of satisfaction achieved justify offering this strategy to families.

**Keywords:** non-ambulant patients; neuromuscular diseases; standing; single-event multilevel surgery; caregiver; satisfaction; function

## 1. Introduction

Some non-ambulatory children with neuromuscular diseases (NMDs), such as cerebral palsy (CP) at Gross Motor Function Classification System level IV, as well as other children with different NMDs who have a similar functional status, are able to stand with assistive devices or the help of another person. Indeed, some can perform sit-to-stand transfers from a wheelchair with assistance. These small functional achievements have a substantial benefit for patients' health [1], and greatly facilitate their care, especially as they enter adulthood and become less manageable for caregivers due to their greater weight.

The proven benefits of standing upright for the physical health of patients include improved bone mineral density; optimization of respiratory capacity; resolution of gastrointestinal issues and spasticity; greater hip stability and range of motion of the hip, knee, and ankle; and better alignment of the spine [1–4]. Standing also has psychological implications, since it enables children to interact more with their surroundings and, by standing at the same height as others, improves their self-esteem [5,6]. When these patients are capable of performing vertical transfers, sometimes bearing part of their weight, this assistance greatly facilitates the process and reduces the physical workload for caregivers, who otherwise tend to develop back pain and other musculoskeletal disorders over time due to the prolonged excessive demands of having to carry children without the autonomy to move [7,8].

Among non-ambulatory children with NMDs, however, rigid ipsilateral hip and knee deformities that develop during growth may eventually limit assisted standing and wheelchair transfers. Other orthopedic abnormalities such as spine deformity merit individual analysis and should be managed by experienced multidisciplinary teams, depending on the magnitude of the curve, extent of cardiorespiratory impairment, and the degree of skeletal maturity [9,10]. Among the lower limb deformities in this subgroup of non-ambulatory children with NMDs, emphasis has usually been placed on the status of the hips and spine to achieve a correct sitting position with a horizontal pelvis and a balanced spine so as to avoid the pain associated with chronically dislocated hip [10]. Despite its relevance to the physical and mental health of children and caregivers, the ability to stand is commonly overlooked, and as a result many patients lose this ability over the course of their lives. Thus, problems in the hips are generally treated individually, while knees and feet are somewhat ignored.

From our clinical perspective, in this subgroup of non-ambulatory children with NMD and progressive impairment in their ability to stand and perform transfers, realignment surgery of the lower limbs, including bifocal (proximal and distal) femoral osteotomies (BFOs), may help patients regain their standing ability by resolving both hip subluxation and ipsilateral flexion deformity of the knee, as well as foot problems that impair weight bearing. Nonetheless, performing single-event multilevel surgery (SEMLS) in these frail patients, including double osteotomies on the same anatomical segment, may increase the surgical time and the complication rate.

Following the Goldberg criteria for measuring outcomes in CP [11], the aims of this study are to analyze the clinical outcomes, safety, and efficacy of SEMLS for lower limb surgery requiring BFOs to restore standing function, and to evaluate patient and caregiver satisfaction with the treatment.

## 2. Materials and Methods

A retrospective observational study was conducted in a pediatric referral center for neuro-orthopedic care, after Institutional Review Board approval (no. R-0071-22). The inclusion criteria were: (1) Children with an NMD and limited ambulation who had previously used a standing frame but later had begun to use the device less often or had abandoned it altogether; and (2) Who had undergone proximal and distal BFOs between 2019 and 2021 to resolve both hip subluxation and ipsilateral knee flexion deformity that limited their ability to stand. Children without head control, those with severe suprapelvic obliquity or a prominently dystonic phenotype, and patients with severe behavioral problems were excluded. All patients were required to have a 2-year minimum follow-up after the intervention.

The surgical approach proposed requires thorough preoperative planning and patient evaluation by a multidisciplinary team comprising pediatric orthopedic surgeons, rehabilitation physicians, physiotherapists, psychologists, anesthesiologists, and pediatricians. The treatment is individualized based on the orthopedic condition of each patient, though all children included in this series had a hip flexion and adduction contracture related to subluxation associated with ipsilateral knee flexion contracture and foot deformities. All in-

terventions were performed by at least 2 orthopedic surgeons experienced in the treatment of NMD and 2 assistant surgeons, in order to optimize surgical time and outcomes [12]. The surgical strategy consists of operating on the affected hip through reconstructive or palliative hip surgery depending on the condition of the femoral head while conducting surgery on the ipsilateral knee to correct the flexion deformity by psoas and rectus tenotomies [13], as well as distal femoral extension osteotomy. Simultaneously, the other surgical team performs contralateral foot surgery or soft-tissue surgery and continues with surgery of the contralateral proximal or distal femur, which is usually necessary due to the underlying deformity or to balance the limb length discrepancy. In the end, the remaining procedures are completed, and the incisions are closed in a coordinated manner between the two surgical teams. A tourniquet is usually reserved for foot surgery. In all cases, a postoperative epidural catheter was used for pain control, knee immobilizers and an abductor triangle were used, as well as short-leg casts if any procedure was performed on the foot. Standing frames were used 4–6 weeks after surgery supervised by physiotherapists, and post-operative ankle foot orthosis tuning was performed by orthopedic technicians with extensive experience in this type of patients. Implant removal surgery, if performed, serves as an opportunity for minor adjustments and tuning.

Demographic data such as age and sex, and clinical information such as type of NMD, functional level measured with the GMFCS, antiepileptic-drug medication, the side affected, the degree of hip and knee flexion deformity, as well as other findings from an orthopedic physical examination, were collected from electronic medical records. Data regarding the surgical procedure, such as date, surgical time, number and description of the procedures performed, number and qualifications of the attending surgeons, and intraoperative incidents were recorded. The following postoperative complications were noted: significant blood loss and need for transfusion, wound dehiscence, superficial or deep infection, uncontrolled pain, nerve or vascular injury, peri-implant fracture, delayed consolidation, and hardware failure, among others.

Postoperatively, the magnitude of hip and knee deformity correction was evaluated, as well as the change in the use of a standing frame and the ability to take transfer steps.

Finally, functional improvements and the degree of satisfaction of the main caregivers were assessed through a simple, brief questionnaire developed by our team and based on the CareQ [14] and the CPCHILD [15] scoring systems. As the questionnaire was specifically designed to detect changes after SEMLS for limb realignment to optimize standing and satisfaction after the procedure, this instrument is fundamentally different from others like it in that it is focused on the functional objective pursued. Patients and caregivers were informed of the study during the final outpatient visit, and caregivers were asked to answer the questionnaire. The questionnaire consists of 18 questions divided into different domains: general health (Q1–Q3), hygiene and basic care (Q4–Q6), positioning and transfers (Q6–Q11), psychological status (Q12–Q13), social aspect (Q14–Q15), and degree of patient or caregiver satisfaction after the procedure (Q16–Q18). Possible responses were: worse, no change, slight improvement, moderate improvement, and significant improvement for questions 1–15; and strongly disagree, disagree, neutral, agree, and strongly agree for questions 16–18. Each response was given a score from 1 to 5, with higher values indicating a more favorable outcome. The value for the last question was multiplied by 3, giving a maximum total score of 100 points (Figure 1).

A descriptive analysis was performed for all variables using Microsoft Excel 2022 (version 16.64; Microsoft Corporation, Redmond, WA, USA). Continuous quantitative data were expressed as median and minimum and maximum values, whereas categorical variables were reported as frequency and percentage values. Differences between preoperative and postoperative scores were analyzed with a Wilcoxon signed-rank test for paired samples. A $p$-value < 0.05 was considered statistically significant and all tests were 2-tailed.

**Questionnaire of Improvements in Daily Life and Satisfaction of Non-Ambulatory Children with NMD and their Caregivers after Multilevel Surgery.**

| *Points* | | 1 | 2 | 3 | 4 | 5 |
|---|---|---|---|---|---|---|
| *What is the degree of change experienced in…?* | | Worse | No change | Slight improvement | Moderate improvement | Significant improvement |
| 1 | *Quality of sleep* | | | | | |
| 2 | *Constipation* | | | | | |
| 3 | *Pain* | | | | | |
| 4 | *Teeth, face, and hair vertical hygiene using bathroom mirror* | | | | | |
| 5 | *Vertical back and lower limbs washing* | | | | | |
| 6 | *Toileting or perineum hygiene* | | | | | |
| 7 | *Orthosis application* | | | | | |
| 8 | *Sitting tolerance* | | | | | |
| 9 | *Standing tolerance* | | | | | |
| 10 | *Home transfers* | | | | | |
| 11 | *School/ public place transfers* | | | | | |
| 12 | *Emotional state (happy or sad)* | | | | | |
| 13 | *Self-esteem* | | | | | |
| 14 | *Moments with family or friends* | | | | | |
| 15 | *Outdoor activities (school, community…)* | | | | | |
| *Direct questions* | | Strongly disagree | Disagree | Neutral | Agree | Strongly agree |
| 16 | *Would undergo the same procedure again?* | | | | | |
| 17 | *Would recommend the intervention to another family in the same situation?* | | | | | |
| 18 | *Are you satisfied with the outcomes? (Points x3)* | | | | | |
| **TOTAL SCORE = __ / 100 points** | | | | | | |

**Figure 1.** Researcher-designed questionnaire to measure changes after surgery according to the functional objective pursued.

## 3. Results

### 3.1. Demographic and Preoperative Clinical Data

Six children presented CP, and there were two patients with myelomeningocele and one with chronic inflammatory demyelinating polyneuropathy. Children with CP were classified as functional level IV in the GMFCS, and the children without the disorder had a similar functional status. Their median age at the time of surgery was 12.86 years (9.33–16.72). Mean follow-up was 27.47 months (24.33–46.9). Table 1 shows the demographic and preoperative clinical and functional data.

**Table 1.** Demographic and preoperative clinical and functional data of the patients in the study.

| Patient | Age (y)/Sex | Diagnosis | Functional Level | Side | Preop Hip Flexion Deformity | Preop Knee Flexion Deformity | Preop Standing with Frame? |
|---|---|---|---|---|---|---|---|
| 1 | 12.6/M | CP, spastic diplegia | IV | Left | 30° | 30° | Yes [†] |
| 2 | 13.8/M | Myelomeningocele, L4 | IV | Left | 30° | 50° | Yes [†] |
| 3 | 11.7/F | Myelomeningocele, L3 | IV | Left | 30° | 30° | No |
| 4 | 9.3/M | CP, spastic tetraplegia | IV | Right | 30° | 30° | No |
| 5 | 16.1/M | CP, spastic diplegia | IV | Right | 15° | 25° | No |
| 6 | 13/F | CP, spastic diplegia | IV | Left | 35° | 40° | No |
| 7 | 11.4/F | CP, spastic tetraplegia | IV | Right | 20° | 30° | No |
| 8 | 12.9/M | CP, spastic tetraplegia | IV | Left | 30° | 30° | No |
| 9 | 16.7/M | Chronic demyelinating polyneuropathy | - | Both | 30° | 35° | No |

Y, years; M, male; F, female; CP, cerebral palsy; GMFCS, Gross Motor Function Classification System; [†] with difficulty and progressive deterioration.

### 3.2. Surgical-Related Data

In all, 1 or 2 proximal and 1 or 2 distal femoral osteotomies were performed, thus indicating that all children received a BFO, associated with surgical procedures at other levels (soft-tissue and bone procedures), as shown in Table 2. Within the proximal osteotomies, there were 2 McHale procedures. In 5 children an acetabular osteotomy was performed (4 Dega and 1 shelf), in 4 an open hip reduction was needed, and 4 children received talonavicular arthrodesis as foot surgery. All procedures had a high surgical burden according to the score described by Nahm et al. [12], with a median of 21 points (18–27). Median surgical time was 4 h 15 min (2 h 35 min–5 h 50 min).

**Table 2.** Surgery-related data and postoperative clinical and functional data of the patients in the study.

| Patient | Femoral Osteotomy | Associated Procedures | Surgical Time (h) | Postop Hip Flexion Deformity | Postop Knee Flexion Deformity | Postop Standing with Frame? | FU (m) |
|---|---|---|---|---|---|---|---|
| 1 | 1 PFO + 2 DFO | Dega + ORH + 2PTA + 2DTO + 2TNA + STS | 5:15 | 0 | 0 | Yes | 25.37 |
| 2 | 1 PFO + 2 DFO | Dega + DTO + TNA + 2CS + Cuboid OT + M1 OT + STS | 4:30 | 0 | 0 | Yes | 36.7 |
| 3 | 2 PFO (MH) + 1 DFO | ORH + DTO + MDTH + ADFH + STS | 4:20 | 0 | 5 | Yes | 46.9 |
| 4 | 2 PFO + 1 DFO | Dega OT + STS | 3:35 | 0 | 0 | Yes | 25.97 |
| 5 | 1 PFO + 2 DFO | ORH + Shelf + 2PTA + DTO + TNA | 5:50 | 0 | 0 | Yes | 39.7 |
| 6 | 1 PFO (MH) + 2 DFO | ORH + 2CS + MT OT + STS | 3:30 | 0 | 0 | Yes | 24.4 |
| 7 | 1 PFO + 2 DFO | PFGG + STS | 2:50 | 0 | 0 | Yes | 38.33 |
| 8 | 1 PFO + 2 DFO | Dega + 2TNA + STS | 4:15 | 15 | 10 | Yes | 24.33 |
| 9 | 1 PFO + 2 DFO | STS | 2:35 | 15 | 10 | Yes | 27.47 |

PFO, proximal femur osteotomy; DFO, distal femur osteotomy; MH, McHale; OT, osteotomy; ORH, open reduction of the hip; PTA, patellar tendon advancement; DTO, distal tibial osteotomy; TNA, talonavicular arthrodesis; STS, soft-tissue surgery; CS, calcaneo-stop; PFGG, proximal femur guided growth; MDTH, medial distal tibial hemiepiphysiodesis; ADFH, anterior distal femoral hemiepiphysiodesis; h, hours; m, months.

### 3.3. Postoperative Clinical and Funcional Outcomes

Hip flexion deformity improved by a median of 30° (15–35), from 30° (15–35) to 0° (0–15), $p = 0.008$, while the median improvement in knee flexion deformity was 30° (20–50), from 30° (25–50) to 0 (0–10), $p = 0.009$. After surgery, all children were able to use a standing frame, which is noteworthy given that previously only two of them could use a standing frame and did so with increasing difficulty. Surgery-related data and postoperative clinical and functional data are displayed in Figure 2 and Table 2.

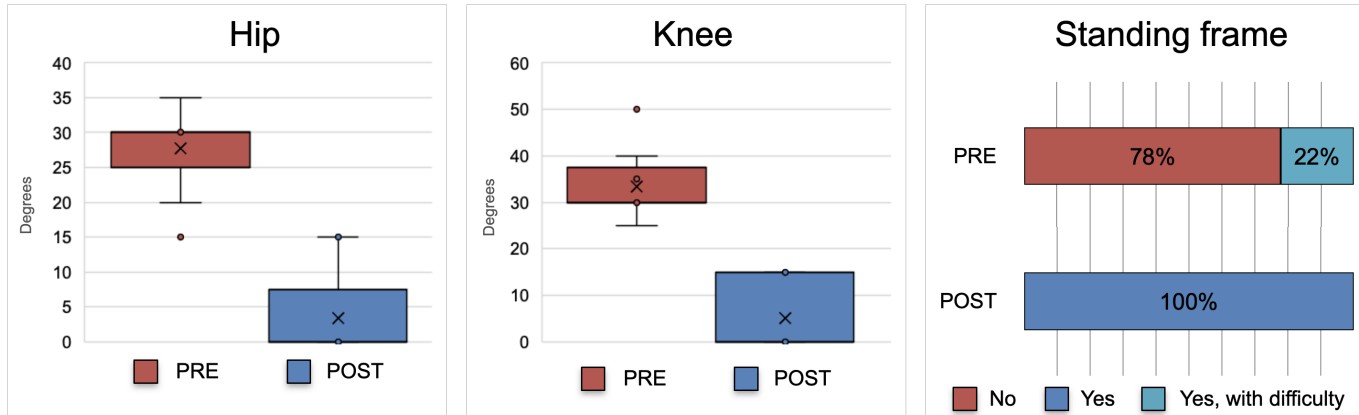

**Figure 2.** Bar plot representing pre- and postoperative values for hip flexion deformity, knee flexion deformity, and standing capacity in a standing frame. Note the improvement in hip and knee flexion contracture as well as the change in the use of a standing frame.

### 3.4. Complications

Among the complications recorded, only moderate blood loss was noted, leading to a median hemoglobin drop of 5.9 (3.8–7.5), with 6 children requiring one transfusion of concentrated red blood cells and 2 children requiring 2 transfusions. Four patients were taking antiepileptic medication (valproate), which is associated with increased bleeding [16]. There were no cases of infection, wound problems, uncontrolled pain, vascular or nerve injury, consolidation problems, or implant failure.

### 3.5. Parents Questionnaire Results

All patients and caregivers reported slight, moderate, or significant improvement in all domains. Specifically, 2/3 or more of patients and caregivers reported moderate or significant improvement for all questions related to positioning and transfers, psychological status, social aspects, and satisfaction (Q7–Q18), as well as for the questions on pain (Q3) and toileting or perineum hygiene (Q6). The aspects that improved the least were quality of sleep; constipation; teeth, face, and hair hygiene; and washing of the back and lower limbs. However, no patients reported worsening in any of the aspects assessed. When caregivers were asked whether they would accept the same intervention again or recommend it to another family in the same situation, all agreed or strongly agreed, and their degree of satisfaction was in most cases high or very high (Figure 3). The median total score of the questionnaire was 83 (63–91) points.

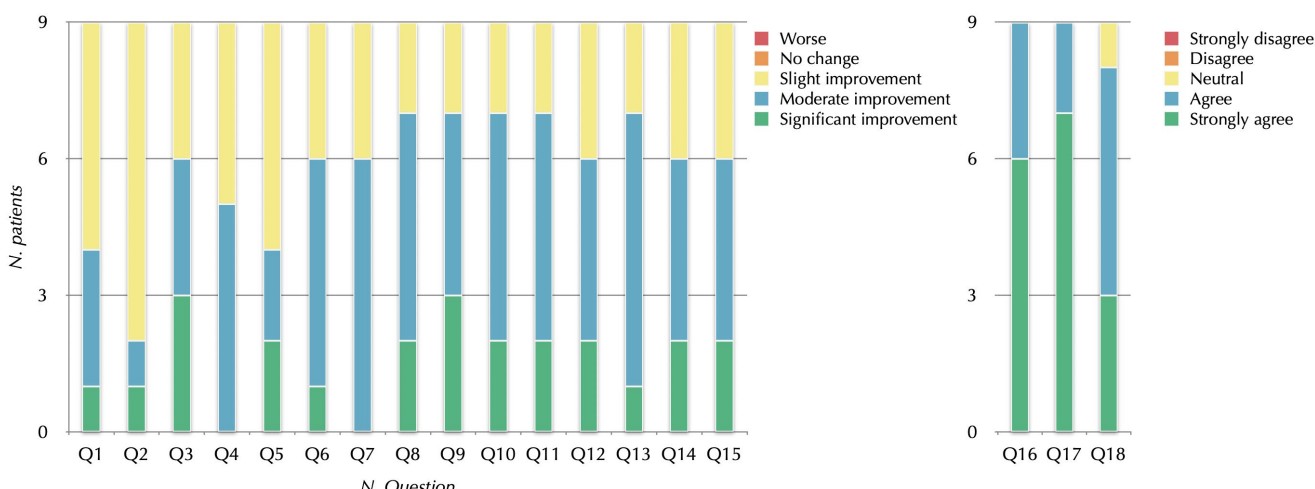

**Figure 3.** Diagram showing the answers of the 9 patients to each of the questionnaire questions.

Some of the caregivers' responses to the question of whether they would undergo the same procedure again were:

- *"Despite the pain after surgery, improving her quality of life has been worth it, she would not have progressed otherwise."*
- *"Before the surgery, my child could not walk and now he can, always with crutches or assistance and for short distances."*

When asked if they would recommend the surgery to another family in the same situation, some of the answers were:

- *"It was a blessing to have had the surgery!"*
- *"Having exhausted other options such as botulinum toxin, orthotics and physical therapy, the surgery has made a total change in his body that allows him to take steps with an adequate posture and without pain."*
- *"My son has substantially improved his quality of life; it has been hard for a year, but it is worth it. If he had not undergone surgery his situation would have been worse."*

Figures 4 and 5 show pre- and postoperative clinical and radiologic images of a few patients in the study.

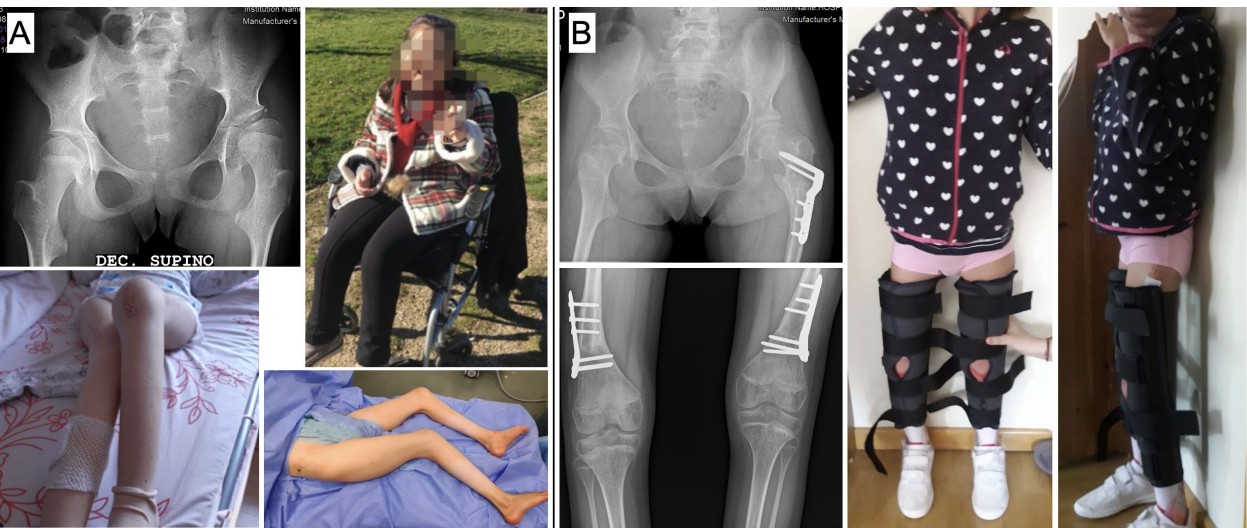

**Figure 4.** Pre- and postoperative clinical and radiological images of a child in the study: (**A**) Preoperative radiographs showing subluxation of the right hip and clinical images showing bilateral knee flexion deformity; (**B**) After proximal femoral varus, derotation, extension and shortening osteotomy combined with distal femoral extension and shortening osteotomy and soft-tissue surgery, the patient can stand independently with the aid of bilateral knee and ankle-foot orthoses.

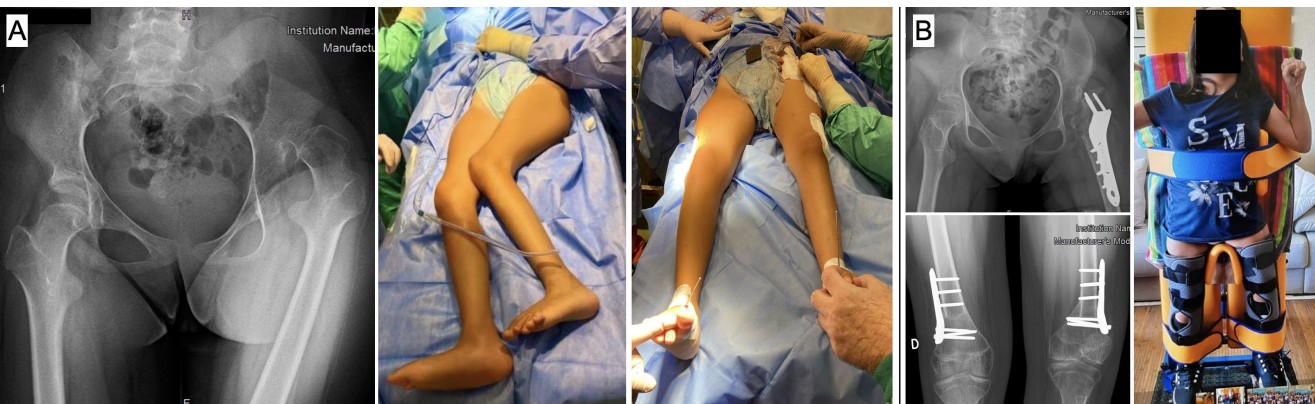

**Figure 5.** Pre- and postoperative clinical and radiologic images of another patient in the study: (**A**) The preoperative radiograph shows complete dislocation of the left hip, with morphologic

deterioration of the femoral head, while the clinical image on the left shows a windswept deformity of the lower limbs, with left knee flexion and length discrepancy; the intraoperative image on the right shows the result of limb realignment, with the dressings covering the wounds on the proximal left thigh and the distal lateral level of both thighs and the wires from the foot surgery; (**B**) Postoperative radiographs show palliative left hip surgery performed with a McHale-type osteotomy and distal femur extension and shortening osteotomies. The clinical image shows the patient erect on a standing frame with her legs well aligned.

## 4. Discussion

SEMLS is widely considered the accepted treatment for improving motor function in ambulatory children with CP (levels I–III) [17–19], and there is also consensus on the surgical indication in non-ambulatory CP children (GMFCS IV–V). The goal of surgical intervention in non-ambulatory children with NMD is to increase hip stability and achieve correct sitting and pain control. However, to date there is little evidence to support or challenge the efficacy of SEMLS in preserving standing ability.

The 2012 study by Blumetti et al. [20] is the only published research that evaluates the results of lower limb SEMLS in patients with level IV CP and suggests that orthopedic surgery is unlikely to maintain or restore mobility. The authors use the Functional Mobility Scale as a postoperative outcome measure, which we believe is inappropriate in these children since the aim of treatment is to enable them to stand rather than walk. Fit-for-purpose measurement tools should be used, as in our study, where we assessed the ability to comfortably stand upright before and after surgery, the ability to perform transfers, and various day-to-day concerns facilitated if the patient can maintain a standing posture, such as constipation; transfers in the home, school, and public places; performing hygiene tasks on the teeth, face, and hair while standing using a bathroom mirror; back and leg washing while upright; as well as psychological well-being, moments with family or friends, and activities outside the home.

Blumetti et al. report a high rate of complications, especially with knee extensor osteotomy, including seven neurapraxias, a complication that can be reduced by shortening the bone [20]. Though the same approach was used in our series, we did not detect any complications of this type. The need to shorten the bone when performing osteotomies to avoid neurovascular complications also functions as an antispastic treatment by reducing muscle tension. Both the benefit of the relative lengthening of the muscles in relation to the bone produced by shortening and the need to avoid substantial length discrepancies between the lower extremities justify the need to act on the contralateral limb. The only other complication recorded in our study was blood loss requiring transfusion, which is consistent with other studies on major orthopedic surgery in patients with CP [16,21]. In this context, in which all our surgeries had a high surgical burden, including technically demanding procedures such as a BFO, and with complications being reasonable and manageable, we consider it rational to contemplate SEMLS in non-ambulatory children, provided the team of surgeons is sufficiently experienced and coordinated. Our message, in contrast to that transmitted by Blumetti et al., is that performing SEMLS in non-ambulatory patients with NMD who have lost the standing ability is effective, safe, and worthwhile for caregivers.

Despite the unfavorable conclusion reached by Blumetti et al., when the caregivers were asked if the surgery was worthwhile or if they would consent to undergo surgery again, 78% answered yes [20]. These results are in line with those obtained in our series, where all patients and/or caregivers responded positively to the questions regarding whether they would undergo the same intervention again or whether they would recommend it to another family in the same situation. We believe this is caused by the fact that the surgery provided better posture for standing and transfers.

Studies on long-term quality of life after hip reconstructive surgery or spinal deformity surgery concluded that surgical treatment significantly improves the quality of life of

children, though it does not alleviate the workload of caregivers [22]. Furthermore, the physical demands on caregivers increase over time as the child develops [8]. We believe that in terms of functionality, achieving assisted standing and transfer ability are two of the factors that most substantially contribute to reducing the burden on caregivers and improving their degree of satisfaction, in addition to improving the quality of life of children, their bone mineral density, gastrointestinal transit, respiratory problems, and self-esteem, among others. For this reason, it is necessary to act not only at the level of the dislocated hip or deformed spine, but also on other levels such as the contralateral hip if the patient presents subluxation, the knees if they exhibit flexion deformity, and the feet if the deformity prevents correct support, in order to realign the lower limbs. This involves performing a BFO in cases of ipsilateral hip and knee deformity, as well as contralateral osteotomies to treat leg length discrepancy in a multilevel lower limb surgery, despite its complexity and aggressiveness for the child.

Prior to such interventions, it is essential to optimize the patients' health and assess patients in a multidisciplinary and integrated approach. This multidisciplinary approach and detailed planning are not limited to the preoperative period, but should cover the entire perioperative and recovery period, during approximately the first year. It is also essential to inform and prepare patients and their caregivers physically and mentally for the lengthy process, and to counsel them on expectations.

This study has some limitations, such as its retrospective nature, small sample size, and the use of an unvalidated assessment questionnaire. This last limitation is justified by the need for a specific questionnaire to detect improvements in the daily life and satisfaction of non-ambulatory children with NMD and their caregivers after SEMLS. Despite the limitations, this study has several strengths that merit highlighting, the main one being that it is the first to evaluate the implementation of BFOs in the context of SEMLS in children who have limited ambulation, with a clear objective of realigning the lower limbs to recover and preserve the standing ability. In addition, outcome measurement tools that are appropriate to the functional objective pursued have been used in the medium term, obtaining favorable results and a high degree of satisfaction from patients and caregivers, which justifies offering this strategy to families.

## 5. Conclusions

In summary, standing position and transfers are extremely important in non-ambulatory children with NMD, and we believe it is our responsibility to ensure that children who still have or have once had these functional capacities do not lose them or can regain them if limitations are due to correctable orthopedic deformities. We must not lose sight of the fact that treatment strategies are changing, that innovation must form part of our agendas, and that these patients will likely benefit from the use of robots and exoskeletons in the future to compensate for their deficient motor control and weakness, allowing them to move autonomously. In this context, the findings of this study suggest that SEMLS is effective, safe, and worthwhile for caregivers and could be considered in non-ambulatory children with NMD to correct hip, knee, and foot deformities and simultaneously realign their lower limbs so as to achieve a functional standing, which can reduce the burden on caregivers and make them potential candidates for rehabilitation robotics and use of exoskeletons [23,24].

**Author Contributions:** Conceptualization: M.G.-O., I.M.-C., G.C.-A., R.M.E.-G. and S.L.-L.; Data curation: M.G.-O., I.M.-C., G.C.-A., R.M.E.-G., D.S.-L., A.R.-B., M.F.-C. and S.L.-L.; Formal analysis: M.G.-O., I.M.-C., G.C.-A., R.M.E.-G., D.S.-L., A.R.-B., M.F.-C. and S.L.-L.; Investigation: M.G.-O., I.M.-C., G.C.-A., R.M.E.-G., D.S.-L., A.R.-B., M.F.-C. and S.L.-L.; Methodology: M.G.-O., I.M.-C., G.C.-A., R.M.E.-G., D.S.-L., A.R.-B., M.F.-C. and S.L.-L.; Project administration: M.G.-O., I.M.-C., G.C.-A., R.M.E.-G. and S.L.-L.; Resources: M.G.-O., I.M.-C., G.C.-A., R.M.E.-G., D.S.-L., A.R.-B., M.F.-C. and S.L.-L.; Supervision: M.G.-O., I.M.-C., G.C.-A., R.M.E.-G., D.S.-L., A.R.-B., M.F.-C. and S.L.-L.; Validation: M.G.-O., I.M.-C., G.C.-A., R.M.E.-G. and S.L.-L.; Visualization: M.G.-O., I.M.-C., G.C.-A., R.M.E.-G., D.S.-L., A.R.-B., M.F.-C. and S.L.-L.; Writing—original draft: M.G.-O. and I.M.-C.; Writing—review and editing: M.G.-O., I.M.-C., G.C.-A., R.M.E.-G., D.S.-L., A.R.-B., M.F.-C. and S.L.-L. All authors have read and agreed to the published version of the manuscript.

**Funding:** This research received no external funding.

**Institutional Review Board Statement:** The study was conducted in accordance with the Declaration of Helsinki and approved by the Institutional Review Board of CEIm Hospital Infantil Universitario Niño Jesús (no. R-0071-22; 28 September 2022) for studies involving humans.

**Informed Consent Statement:** Informed consent was obtained from all subjects involved in the study.

**Data Availability Statement:** The data presented in this study are available on request from the corresponding author.

**Acknowledgments:** We would like to express our gratitude to the participants and their families for their invaluable help in this study.

**Conflicts of Interest:** The authors declare no conflict of interest. The funders had no role in the design of the study; in the collection, analyses, or interpretation of data; in the writing of the manuscript; or in the decision to publish the results.

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
