# Peer review of "Can We Extend the Indications for Multilevel Surgery to Non-Ambulatory Children with Neuromuscular Diseases? A Safety and Efficacy Study"

_2673-4095, doi:10.3390/surgeries4020022_

Round 1
Reviewer 1 Report
This paper reported surgical outcomes of SEMLS for CP patient.
The number of patients were only 9 patients with 2 years follow-up period. Surgical procedures were common not innovative. There was no interesting results and findings in this paper.
Reviewer 2 Report
I read with a great interest the manuscript entitled “Can we extend the indications for multilevel surgery to non-ambulatory children with neuromuscular diseases? A safety and efficacy study” written by Galán-Olleros et al.
It is a retrospective, descriptive study aiming to o analyze the safety and efficacy of single-event multilevel surgery (SEMLS) involving bifocal femoral osteotomy performed in 9 non-ambulatory children around the age of 12 with neuromuscular diseases to resolve both hip subluxation and ipsilateral knee flexion contracture that impaired standing, and to evaluate patient and caregiver satisfaction.
The authors provide data to suggest that the above interventions can safely improve functional standing, wheelchair transfer and quality of life of children and caregivers and justify offering this option to families.
The topic is appealing, innovative and the manuscript is thorough and engaging from the first read. The objective is clearly stated and the quality of English language used is good. Overall, the methodology is excellent, well-structured and the results presented in detail.
No plagiarism, fraud or ethical concerns have been raised.
Reviewer 3 Report
The manuscript is interesting, I have a few comments:
1. Lines 94-115: Is this description of what you did or general description of surgery in such patients? It is not clear due to the text in present time.
2. Figure 1 should be moved to supplementary material.
3. Please perform also statistical analysis of the data, especially comparing pre and postoperative values in Figure 2.
4. This study is in essence just a case report series of 9 patients, without hypothesis being tested and without real aim; therefore, I believe it should be designated that it is a case report. The following would also properly designate the level of evidence these results offer.
5. “This study has some limitations, such as its retrospective nature, small sample size, and the use of an unvalidated assessment questionnaire.” These are very severe limitations. Furthermore, the study did not have any controls.
6. Line 305: Please change “should” to “could”, since there are actually no real evidence that support your claim.
Reviewer 4 Report
Abstract
1. Remove "Methods:" from the beginning of the abstract. The abstract should start with the study's aim.
2. Use a more precise term than "non-ambulatory children" to describe the patient population, such as "non-ambulatory children with neuromuscular diseases (NMD)".
3. For the surgical time range "(2h35m-5h50m)", use a consistent format: "(2h35min-5h50min)".
4. Add a comma after "Mean follow-up was 27.47 months (24.33-46.9)" to separate it from the next sentence.
5. Revising the sentence as "as nearly all (8/9) were very satisfied".
Introduction
The introduction is ok. Well written with sufficient background work.
Methods
1. In line 97, "through" should be "though".
Results
Figures are ok. Results and tables are clearly presented.
Discussion
Very well written
Conclusion
I would suggest not to use citations in the conclusion section
References
They are ok
Reviewer 5 Report
The topic of this manuscript is interesting and fits well the scope of Surgeries. The reviewer feels it can only be accepted after some necessary amendments.
(1) This study is based the results from less than 10 patients. Therefore, the conclusion should be tuned down and the findings SUGGEST ...
(2) The gender and age of the patients should be indicated in Table 2.
Round 2
Reviewer 1 Report
As previously mentioned, mutilevel surgeries for CP patients are very common.
Simultaneous surgeries for both knee and hip are also possible in nonambulatory CP patients.
I agree that double osteotomies for femur is challenging in nonambulatory CP patients.
Reviewer 3 Report
A stated previously, the manuscript is interesting and has merit, however it is a case report series of 9 patients, without hypothesis being tested, therefore I believe it should be designated that it is a case report. The authors satisfactorily addressed my other comments.
